# Effect of Horizontal Quasi-Periodic Oscillation on the Interfacial Instability of Two Superimposed Viscous Fluid Layers in a Vertical Hele-Shaw Cell

Mouh Assoul [1], Abedelouahab El Jaouahiry [1], Jamila Bouchgl [2], Mourad Echchadli [1] and Saïd Aniss [1,*]

1   Laboratory of Mechanics, Faculty of Sciences Aïn Chock, University Hassan II-Casablanca, Casablanca 20100, Morocco
2   Higher Institute of Marine Fisheries, Agadir 80000, Morocco
*   Correspondence: said.aniss@etu.univh2c.ma

**Abstract:** We investigate the effect of horizontal quasi-periodic oscillation on the stability of two superimposed immiscible fluid layers confined in a horizontal Hele-Shaw cell. To approximate real oscillations, a quasi-periodic oscillation with two incommensurate frequencies is considered. Thus, the linear stability analysis leads to a quasi-periodic oscillator, with damping, which describes the evolution of the amplitude of the interface. Two types of quasi-periodic instabilities occur: the low-wavenumber Kelvin-Helmholtz instability and the large-wavenumber resonances. We mainly show that, for equal amplitudes of the superimposed accelerations, and for a low irrational frequency ratio, there is competition between several resonance modes allowing a very large selection of the wavenumber from lower to higher values. This is a way to control the sizes of the waves. Furthermore, increasing the frequency ratio has a stabilizing effect for both types of instability whose thresholds are found to correspond to quasi-periodic solutions using the frequency spectrum. For a ratio of the two superimposed displacement amplitudes equal to unity and less than unity, the number of resonances and competition between their modes also become significant for the intermediate values of the ratio of frequencies. The effects of other physical and geometrical parameters, such as the damping coefficient, density ratio, and heights of the two fluid layers, are also examined.

**Keywords:** Hele-Shaw cell; quasiperiodic oscillation; Kelvin-Helmholtz and parametric instabilities

## 1. Introduction

The instability of the interface of two superimposed immiscible liquid layers subjected to periodic horizontal oscillation is of practical importance and was studied by many researchers in various configurations [1–14]. Kelly [1] investigated the Kelvin-Helmholtz instability with two oscillating flows where the velocity fields are periodic. Using the linear stability analysis, he discovered situations where the oscillation stabilizes the unstable shear flow. Afterwards, Wolf [2,3] carried out the first experiments in which he observed the formation of a quasi-steady relief, called a frozen wave. Note that this relief develops when the oscillation intensity exceeds a specific threshold. Subsequently, the interface behavior of the fluid layers of different densities was theoretically studied by Lyubimov et al. [4]. They reduced the linear stability problem under the inviscid approximation to the Mathieu equation. It was found that the basic instability mode, associated with the development of the Kelvin-Helmholtz instability at the interface between counter-streaming flows, occurs, and that the possibility of parametric resonance can take place. For inviscid fluids, there is no threshold for the parametric instability. Furthermore, for sufficiently high vibration frequencies, the parametric instability is sensitive to viscous damping and thus occurs over a narrow range of wavenumbers. The work by Lyubimov et al. [4] was extended by Khenner et al. [5] to the parametric resonant regions of instability associated with capillary wave intensification at the interface. Note that, in these works [4,5], numerical

results were also discussed in the viscous case. It was found that the parametric instability is strongly damped by viscosity and the Kelvin-Helmholtz mode is the most unstable. Later, Ivanova et al. [6] experimentally found that, in the circular translational oscillation of a system of immiscible fluids of different densities, a steady-state relief excitation, associated with the Kelvin-Helmholtz instability, is determined by the square of vibrational velocity, while the azimuthal fluid flow associated with the generation of an average vorticity in the skin-layers is determined by the oscillatory Reynolds number. Subsequently, Talib et al. [7,8] experimentally and numerically investigated the effect the viscosity contrast on the linear stability threshold using the spectral collocation method. They showed that, depending on the value of the density contrast, either the Kelvin–Helmholtz or the first resonant mode may be the most unstable. These two modes exhibit opposite dependencies on the viscosity contrast with a sharp stabilization of the first resonant mode, while the threshold of the Kelvin–Helmholtz mode exhibits a sharp reduction. Thereafter, Yoshikawa et al. [9] conducted a theoretical study to investigate the effect of viscosity and its contrast at the interface. This study was considered for the liquid layers of infinite depth and for an amplitude of the oscillation smaller than the wavelength of the perturbation. The results in this case are in good agreement with those of the previous experiments and theoretical studies [7,8]. Experiments were also conducted by Yoshikawa et al. [10] to show that the threshold and wavenumber significantly depend on the oscillation frequency. It turns out that the waves found are longer than those predicted by the inviscid theory of the Kelvin-Helmholtz oscillatory instability. In the same spirit, Jalikop et al. [11] showed that the gravity-capillary waves in a horizontally oscillating two-layer system, which are often referred to as frozen waves, lose stability in the presence of oscillatory transverse amplitude modulation.

A few research groups examined theoretically [12,13] and experimentally [14] the effect of periodic horizontal and vertical oscillations on the interfacial instability between two viscous superimposed immiscible fluid layers contained in a vertical Hele-Shaw cell. For instance, Bouchgl et al. [12] performed an inviscid linear stability analysis of the viscous basic flow leading to the periodic Mathieu oscillator describing the evolution of the interface amplitude. They showed that a decrease in the viscosity contrast has a stabilizing effect on the Kelvin-Helmholtz instability, which is displaced towards the long-wave region. Hereafter, Lyubimova et al. [13] extended the investigation by Bouchgl et al. [12] by taking into account the viscosity in the perturbed equations. They discussed the influence of different physical parameters on the stability of the interface. Recently, and on the experimental side, Li et al. [14] investigated an extreme case of two coupled Faraday waves of three layers in a covered Hele-Shaw cell with periodic vertical vibration. More recently, the works by Bouchgl et al. [12] and that by Lyubimova et al. [13] were extended to a fully saturated porous media [15]. Bouchgl and Aniss [15] showed that the Darcy number has a destabilizing effect on the parametric instability and on the Kelvin-Helmholtz instability. Furthermore, the decrease in permeability significantly increases the stability threshold of the parametric instability, which is displaced to the short-wave regions.

When there is a large difference in velocity between the two fluid layers, Kelvin-Helmholtz type instabilities develop at the interface. Their amplitudes increase and when they reach a critical value, the crest of the wave is torn off, giving rise to a fragment whose size would be proportional to the wavelength of these waves. This type of mechanism is used in the study of the fragmentation of liquid jets. Our motivation concerns the control of the wavelength by considering the Kelvin-Helmholtz instability in the presence of real oscillations which generally have several frequencies in contrast to the previous works using the standard periodic oscillation. Let us note, on the one hand, that the different frequencies are, in general, incommensurable with each other, so that their ratios are irrational numbers. On the other hand, the instability occurs in the form of the standard Kelvin-Helmholtz instability as well as in the form of parametric resonances. Inspired by the work of Rand et al. [16] who studied the quasi-periodic Mathieu equation, Boulal et al. [17,18] used this type of modulation in Rayleigh–Bénard convection where it is shown that the ratio of frequencies allows one to control the convection threshold. This type of modulation was

also used by Yagoubi et al. [19] to study the effect of vertical quasi-periodic oscillation on the stability of the free surface of a horizontal liquid layer. In the present study, oscillation with two incommensurate frequencies is considered and its effect on the interface instability between two superimposed layers of viscous immiscible fluids confined in a Hele-Shaw cell is examined.

The paper is structured as follows. In Section 2, we determine the basic flow considered as quasi-periodic and viscous. After that, we perform in the same section a linear stability analysis in which the governing equations are reduced to a quasi-periodic oscillator. In Section 3, the numerical procedure is explained. In Section 4, the numerical results are presented and discussed. Section 5 is devoted to the conclusions.

## 2. Formulation

### 2.1. Governing Equations

Consider two superimposed viscous fluid layers filling a vertical Hele-Shaw cell of infinite extent in the $x$ direction (see Figure 1). We denote by $h = h_1 + h_2$ the height of the cell, by $e$ the distance between the vertical walls where $\frac{e}{h} \ll 1$; the vertical and horizontal walls are located, respectively, at $z^* = \pm e/2$ and $y^* = -h_1, h_2$.

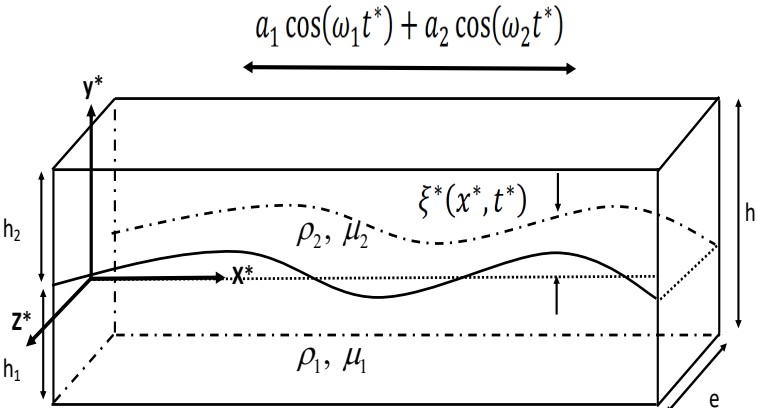

**Figure 1.** Superimposed viscous fluid layers filling a vertical Hele-Shaw cell subjected to quasi-periodic horizontal oscillation.

We assume that the denser fluid occupies the bottom region, of height $h_1$, and the light one occupies the upper region, of height $h_2$, so that the configuration is gravitationally stable. The Hele-Shaw cell is assumed to oscillate according to the law of displacement, $[a_1 \cos(\omega_1 t^*) + a_2 \cos(\omega_2 t^*)]\mathbf{x}^*$, where $\omega_1$ and $\omega_2$ are two-dimensional incommensurate frequencies and $t^*$ is the time. The parameters $a_1$ and $a_2$ are the amplitudes of motion and $\mathbf{x}^*$ is the horizontal unit vector. Therefore, the fluid layers are submitted to the gravitational force field, $\rho_j \mathbf{g}$, and the quasi-periodic force field, $-\rho_j [a_1 \omega_1^2 \cos(\omega_1 t^*) + a_2 \omega_1^2 \cos(\omega_1 t^*)]\mathbf{x}^*$. Note that each fluid layer is characterized by its density $\rho_j$, and its dynamic viscosity $\mu_j$ ($j = 1, 2$). The surface tension at the interface is denoted by $\gamma$. The flows in each fluid layer are governed by the momentum and continuity equations written in the relative frame:

$$\nabla \cdot \mathbf{V}_j^* = 0 \qquad \text{(j=1,2)} \tag{1}$$

$$\frac{\partial \mathbf{V}_j^*}{\partial t^*} + (\mathbf{V}_j^* \cdot \nabla)\mathbf{V}_j^* = -\frac{1}{\rho_j}\nabla P_j^* + \nu_j \Delta \mathbf{V}_j^* - g\mathbf{y}^*$$
$$+ \left[a_1 \omega_1^2 \cos(\omega_1 t^*) + a_2 \omega_2^2 \cos(\omega_2 t^*)\right]\mathbf{x}^* \tag{2}$$

where $\mathbf{V}_j^* = (u_j^*, v_j^*)$ is the dimensional velocity and $P_j^*$ is the hydrodynamic pressure in each fluid layer. The last term of the Navier–Stokes equation (2) represents the inertial

driving forces [20] due to the oscillation. Note that, for a motionless Hele-Shaw cell, Gondret and Rabaud [21] averaged the Navier–Stokes equations with respect to the variable $z^*$ by considering the parabolic velocity field, assuming that there is no transverse velocity and that the two second derivatives in $x^*$ and $y^*$ are negligible with respect to the $z^*$ derivative. Following this approach, the parabolic velocity field is given by:

$$\mathbf{V}_j^*(x^*, y^*, z^*) = \frac{3}{2}\overline{\mathbf{V}}_j^*(x^*, y^*)\left[1 - (\frac{2z^*}{e})^2\right] \tag{3}$$

According to the above assumptions, the averaged momentum equation, with respect to the spatial variable $z^*$, between $-\frac{e}{2}$ and $\frac{e}{2}$, is given by:

$$\frac{\partial \overline{\mathbf{V}}_j^*}{\partial t^*} + \frac{6}{5}(\overline{\mathbf{V}}_j^*.\nabla)\overline{\mathbf{V}}_j^* = -\frac{1}{\rho_j}\nabla P_j^* - \frac{12\mu_j}{e^2}\overline{\mathbf{V}}_j^* - g\mathbf{y}^*$$
$$+ \left[a_1\omega_1^2\cos(\omega_1 t^*) + a_2\omega_2^2\cos(\omega_2 t^*)\right]\mathbf{x}^* \tag{4}$$

Thereafter, we perform a dimensional analysis by means of an appropriate choice of scales used in interfacial instability problems in Hele-Shaw cell [12,13]. Thus, the length is scaled by the capillary length $l_c = \sqrt{\frac{\gamma}{g(\rho_1 - \rho_2)}}$, the time by $\omega_1^{-1}$, the velocity by $a_1\omega_1$, and the pressure by $\rho_1 a_1\omega_1^2 l_c$. Hence, Equation (4) for each fluid layer is written in the dimensionless form as follows:

$$\frac{\partial \overline{\mathbf{V}}_1}{\partial t} + \frac{6}{5}q(\overline{\mathbf{V}}_1 \cdot \nabla)\overline{\mathbf{V}}_1 = -\nabla P_1 - 6\frac{F}{\sqrt{We}}\overline{\mathbf{V}}_1 \tag{5}$$
$$+ \left[\cos(t) + A\Omega^2\cos(\Omega t)\right]\mathbf{x} - \frac{1}{qWe}\mathbf{y}$$

$$\frac{\partial \overline{\mathbf{V}}_2}{\partial t} + \frac{6}{5}q(\overline{\mathbf{V}}_2.\nabla)\overline{\mathbf{V}}_2 = -\frac{1}{\rho}\nabla P_2 - 6\frac{\mu}{\rho}\frac{F}{\sqrt{We}}\overline{\mathbf{V}}_2 \tag{6}$$
$$+ [\cos(t) + A\Omega^2\cos(\Omega t)]\mathbf{x} - \frac{1}{qWe}\mathbf{y}$$

where $q = \frac{a_1}{l_c}$ is the dimensionless amplitude of oscillation, $A = \frac{a_2}{a_1}$ is the amplitude ratio of displacements, $\Omega = \frac{\omega_1}{\omega_2}$ is the irrational ratio of frequencies, $F = \frac{\sqrt{\frac{l_c}{g}}}{\sigma_2}$ is the damping coefficient responsible of the friction, in which $\sigma_2 = \frac{we^2}{2\nu_2}$ is the frequency number ($\nu_2$ is the kinematic viscosity of the upper fluid), $We = \frac{\omega_1^2 l_c}{g}$ is the Weber number, $\rho = \frac{\rho_2}{\rho_1}$ is the density contrast, and $\mu = \frac{\mu_2}{\mu_1}$ is the viscosity contrast.

### 2.2. Base Flows

Due to the horizontal oscillation, it is evident that the base flow in each fluid layer has a one-component velocity field, $\overline{\mathbf{V}}_j^b(t) = (U_j^b(t), 0, 0)$, which is quasi-periodic and parallel to the $x$ axis. In this equilibrium state, the interface between the two fluid layers is considered planar, horizontal, and coincident to the $y = 0$ plane. These flows satisfy the equation of continuity, Equation (1), and the momentum Equations (6) and (7) as well as the condition of flow closeness, as expressed by the counter-flowing layers and given by the additional integral condition expressing the balance of the displacement volume of both fluids [5,13] imposed as below:

$$\int_{-H_1}^0 \overline{\mathbf{V}}_1^b.\mathbf{x}\,dy = -\int_0^{H_2} \overline{\mathbf{V}}_2^b.\mathbf{x}\,dy \tag{7}$$

This counter-flowing is the result of the vertical end walls of the Hele-Shaw cell located at $x = 0, L$, and appears when the vertical volume flow is conserved. Subsequently, the quasi-periodic base velocity field is sought in the form:

$$U_j^b(t) = \alpha_j \cos(t) + \beta_j \sin(t) + \lambda_j \cos(\Omega t) + \Lambda_j \sin(\Omega t) \tag{8}$$

The coefficients $\alpha_j$, $\beta_j$, $\lambda_j$ and $\Lambda_j$ are expressed as a function of the different physical parameters of the problem, as defined in Section 2.1, and are given by:

$$
\begin{aligned}
\alpha_2 &= -\frac{1}{H}\alpha_1, \\
&= \frac{-6F(H+\mu)(1-\rho)}{\sqrt{We}\left[(H+\rho)^2 + 36\left(\frac{F}{\sqrt{We}}\right)^2(H+\mu)^2\right]}, \quad H = \frac{H_2}{H_1}
\end{aligned} \tag{9}
$$

$$
\begin{aligned}
\beta_2 &= -\frac{1}{H}\beta_1 \\
&= \frac{-(H+\rho)(1-\rho)}{(H+\rho)^2 + 36\left(\frac{F}{\sqrt{We}}\right)^2(H+\mu)^2}
\end{aligned} \tag{10}
$$

$$
\lambda_2 = -\frac{1}{H}\lambda_1 \tag{11}
$$

$$
= \frac{-6A\Omega^3 F(H+\mu)(1-\rho)}{\Omega\sqrt{We}\left[\Omega^2(H+\rho)^2 + 36\left(\frac{F}{\sqrt{We}}\right)^2(H+\mu)^2\right]}
$$

$$
\begin{aligned}
\Lambda_2 &= -\frac{1}{H}\Lambda_1 \\
&= \frac{-A\Omega^3(H+\rho)(1-\rho)}{\Omega^2(H+\rho)^2 + 36\left(\frac{F}{\sqrt{We}}\right)^2(H+\mu)^2}
\end{aligned} \tag{12}
$$

Furthermore, the pressure at the equilibrium is given by:

$$P_1^b = -\frac{1}{qWe}y + f(x,t) \tag{13}$$

$$P_2^b = -\frac{\rho}{qWe}y + f(x,t) \tag{14}$$

where $f(x,t)$ is an arbitrary function. Note that, in the limit of a high frequency number, $\sigma_2 \to \infty$ corresponding to $\nu_2 \to 0$, the base-flow solution determined in this work, and given by Equation (8), tends towards the solution corresponding to the oscillating flows in an approximation of the two inviscid-layer system given by Khenner et al. [5].

### 2.3. Linear Stability

The perturbed state, in terms of velocity and pressure, is written as:

$$\overline{\mathbf{V}}_j = \overline{\mathbf{V}}_j^b + \mathbf{v_j}\big(u_j(x,y,t), v_j(x,y,t)\big), \quad P_j^* = P_j^b + p_j(x,y,t) \tag{15}$$

The dimensionless linear system of the conservation equations, for each fluid layer, is given by:

$$\frac{\partial u_j}{\partial x} + \frac{\partial v_j}{\partial y} = 0 \tag{16}$$

$$\rho^{j-1}\left[\frac{\partial u_j}{\partial t} + \frac{6}{5}qU_j^b \cdot \frac{\partial u_j}{\partial x}\right] = -\frac{\partial p_j}{\partial x} - 6\frac{\mu^{j-1}F}{\sqrt{We}}u_j \tag{17}$$

$$\rho^{j-1}\left[\frac{\partial v_j}{\partial t} + \frac{6}{5}qU_j^b \cdot \frac{\partial v_j}{\partial x}\right] = -\frac{\partial p_j}{\partial y} - 6\frac{\mu^{j-1}F}{\sqrt{We}}v_j \tag{18}$$

where $j = 1$ for the lower layer and $j = 2$ for the upper one. Thereafter, the interface is described by the equation $y = \xi(x, t)$, where $\xi(x, t)$ is an infinitesimal perturbation of the base interface. Hereafter, the solution of the system of Equations (16)–(18) is sought in terms of normal modes:

$$\left[ p_j, u_j, v_j \right] = \left[ \tilde{p}_j(t, y), \tilde{u}_j(t, y), \tilde{v}_j(t, y) \right] e^{ikx} \tag{19}$$

$$\xi(t, x) = \tilde{\xi}(t) e^{ikx} \tag{20}$$

where $i^2 = -1$ and $k$ is the wavenumber in the $x$ direction. Thereafter, the velocity potentials, $\phi_j(t, x, y) = \tilde{\phi}_j(t, y) e^{ikx}$, are inserted into the continuity Equation (16), to obtain:

$$\phi_j(t, x, y) = \left[ C_j^1(t) e^{ky} + C_j^2(t) e^{-ky} \right] e^{ikx} \tag{21}$$

Using the linearized kinematic condition,

$$\frac{d\tilde{\xi}}{dt}(t) + ikq U_j^b \tilde{\xi}(t) = q \frac{\partial \tilde{\phi}_j}{\partial y}(t, y) \tag{22}$$

and the slip boundary conditions at the horizontal walls, $\tilde{v}_j(t, y) = 0$ at $y = -h_1$ and $y = h_2$, the constants $C_j^1$ and $C_j^2$ are determined. Thus, the velocity potential for each fluid layer is given by:

$$\tilde{\phi}_1(t, y) = \frac{e^{2kh_1} e^{ky} + e^{-ky}}{qk(e^{2kh_1} - 1)} \left[ \frac{d\tilde{\xi}(t)}{dt} + ikq U_1^b \tilde{\xi}(t) \right] \tag{23}$$

$$\tilde{\phi}_2(t, y) = \frac{e^{-2kh_2} e^{ky} + e^{-ky}}{qk(e^{-2kh_2} - 1)} \left[ \frac{d\tilde{\xi}(t)}{dt} + ikq U_2^b \tilde{\xi}(t) \right] \tag{24}$$

To complete the mathematical formulation, the normal stress balance at the interface is written as:

$$(P_1^b + p_1) - (P_2^b + p_2) = \gamma \nabla \cdot \mathbf{n} \tag{25}$$

where $\gamma$ is the surface tension and $\mathbf{n}$ is the unit vector normal to the interface. Thus, the linearized form of the curvature is $\nabla \cdot \mathbf{n} = -\frac{\partial^2 \xi}{\partial x^2}(x, t) = k^2 \tilde{\xi}(t) e^{ikx}$. Hereafter, to linearize the dynamic condition (25), the total pressure is expressed near $y = 0$ with a first-order Taylor expansion:

$$P_j^b + p_j = P_j^b(0) + \frac{\partial P_j^b}{\partial y} \Big|_{y=0} \xi(t) + p_j(t, x, 0) \tag{26}$$

The systems of Equations (16)–(18) and (23)–(26) allow us to obtain the expression of the damped quasi-periodic oscillator:

$$
\frac{d^2 \tilde{\xi}}{dt^2} + \left[ i \frac{11}{5} \frac{R_1 U_1^b(t) + \rho R_2 U_2^b(t)}{R_1 + \rho R_2} + \frac{6F}{\sqrt{We}} \frac{R_1 + \mu R_2}{R_1 + \rho R_2} \right] \frac{d\tilde{\xi}}{dt}
$$
$$
+ \left[ iqk \left( \frac{R_1 \frac{dU_1^b}{dt}(t) + \rho R_2 \frac{dU_2^b}{dt}(t)}{R_1 + \rho R_2} + \frac{6F}{\sqrt{We}} \frac{R_1 U_1^b(t) + \mu R_2 U_1^b(t)}{R_1 + \rho R_2} \right) \right.
$$
$$
\left. - 6q^2 k^2 \frac{R_1 (U_1^b(t))^2 + \rho R_2 (U_2^b(t))^2}{R_1 + \rho R_2} + \frac{k(1 - \rho)(1 + k^2)}{We(R_1 + \rho R_2)} \right] \tilde{\xi} = 0 \tag{27}
$$

with $R_1 = \coth(kH_1)$ and $R_2 = \coth(kH_2)$. The parametric differential Equation (27) governs the evolution of the interface displacement from its equilibrium state.

## 3. Numerical Procedure

To numerically solve Equation (27), we introduce the following changes in variables:

$$\Omega = \frac{p}{m}, \qquad \tau = \frac{t}{m}, \qquad \frac{\partial}{\partial t} = \frac{1}{m}\frac{\partial}{\partial \tau}, \qquad \frac{\partial^2}{\partial t^2} = \frac{1}{m^2}\frac{\partial^2}{\partial \tau^2} \tag{28}$$

Here, $\Omega$, intervening in the base velocity (8), is approximated by a rational frequency ratio, where $p$ and $m$ are prime numbers between them. For instance, $\sqrt{2} = \frac{1393}{985}$, $\sqrt{3} = \frac{1351}{780}$, $\sqrt{5} = \frac{2889}{1292} \dots \sqrt{37} = \frac{882}{145}$. Inserting these changes of variables into Equation (27), we obtain a periodic Mathieu-equation with a damping term. The Equation (27) can be converted into the state space form by considering the system states $[x_1, x_2]^T$ where $x_1 = \tilde{\zeta}$ and $\frac{dx_1}{d\tau} = x_2$. Thus, Equation (27) is written in a matrix form:

$$\left\{ \begin{array}{c} \frac{dx_1}{d\tau} \\ \frac{dx_2}{d\tau} \end{array} \right\} = \left[ \begin{array}{cc} 0 & 1 \\ -m^2\alpha(\tau) & -m\beta(\tau) \end{array} \right] \left\{ \begin{array}{c} x_1 \\ x_2 \end{array} \right\} \tag{29}$$

where:

$\alpha(\tau) = i\frac{11}{5}\frac{R_1 U_1^b(\tau) + \rho R_2 U_2^b(\tau)}{R_1 + \rho R_2} + \frac{6F}{\sqrt{We}}\frac{R_1 + \mu R_2}{R_1 + \rho R_2}$

$\beta(\tau) = iqk\left( \frac{R_1 \frac{dU_1^b}{dt}(\tau) + \rho R_2 \frac{dU_2^b}{dt}(\tau)}{R_1 + \rho R_2} + \frac{6F}{\sqrt{We}}\frac{R_1 U_1^b(\tau) + \mu R_2 U_1^b(\tau)}{R_1 + \rho R_2} \right) - 6q^2 k^2 \frac{R_1 (U_1^b(\tau))^2 + \rho R_2 (U_2^b(\tau))^2}{R_1 + \rho R_2}$

$+ \frac{k(1-\rho)(1+k^2)}{We(R_1 + \rho R_2)}$

The Floquet theory [22] is applied to matrix system (29). This theory then states that there exists a constant matrix $\mathbf{R}$ such that:

$$\mathbf{S}(\tau + T) = \mathbf{R}\mathbf{S}(\tau) \tag{30}$$

where $T$ is the period and $\mathbf{S}$ is the fundamental solution matrix of the system (29) satisfying:

$$\frac{\partial \mathbf{S}}{\partial t} = \left[ \begin{array}{cc} 0 & 1 \\ -m^2\alpha(\tau) & -m\beta(\tau) \end{array} \right] \mathbf{S} \tag{31}$$

In addition, if the eigenvalues of the matrix $\mathbf{R}$ are $\gamma_j$ ( $j = 1, 2$), then the solution of system (29) can be written as:

$$x_j = Z_j(\tau)\exp(\lambda_j) \tag{32}$$

where $\mathbf{Z}$ is a periodic function of period $T$ and the coefficients $\lambda_j$ are the Floquet exponents that are related to the eigenvalues $\gamma_j$ by:

$$\lambda_j = \frac{1}{T}\ln(\gamma_j) \tag{33}$$

To calculate $\lambda_j$, we first determine the matrix $\mathbf{R}$ using the relation (30) to obtain:

$$\mathbf{S}(T) = \mathbf{R}\,\mathbf{S}(0) \tag{34}$$

Thus, to calculate $\mathbf{R}$, we use a fourth-order Runge–Kutta numerical scheme for the integration of system (31) over one period with the initial condition $\mathbf{S}(0) = \mathbf{I}$, where $\mathbf{I}$ is the identity matrix. Once the eigenvalues of the matrix $\mathbf{R}$, $\gamma_j$, are determined, we calculate the Floquet exponents $\lambda_j$ using Equation (33). Thereafter, we only consider the most unstable mode corresponding to the Floquet exponent with the largest real part denoted by $\lambda_r$ (temporal growth rate). If $\lambda_r > 0$, the system is unstable, and if $\lambda_r < 0$, the system is stable.

## 4. Results and Discussion

In this investigation, the marginal stability curves corresponding to the variation in the dimensionless amplitude of oscillation, $q$, versus the wave number, $k$, are numerically determined for assigned values of the irrational ratio of frequencies, $\Omega = \frac{\omega_1}{\omega_2}$; the ratio of amplitudes of oscillation, $A = \frac{a_2}{a_1}$; the coefficient of friction, $F$; the Weber number, $We$; the density contrast, $\rho$; the viscosity contrast, $\mu$; and the layer depths $H_1$ and $H_2$.

### 4.1. Validation of the Numerical Procedure

The numerical procedure is validated in the case of periodic oscillation and the obtained results are compared to those of Lyubimova et al. [13]. Figure 2 illustrates the marginal stability curves in the case of periodic oscillation [5] corresponding to $A = \frac{a_2}{a_1} = 0$ for the representative values $We = 5$, $F = 0.1$, $\mu = 0.5$, $\rho = 0.8$, and $H_1 = H_2 = 2$. It is worth noting that the results converge, in excellent agreement, towards those by Lyubimova et al. [13]. As indicated in previous works [5,7,8,12,13], two types of instabilities occur at the interface, the Kelvin-Helmholtz instability at low wavenumbers and the parametric one (resonance) at high wavenumbers.

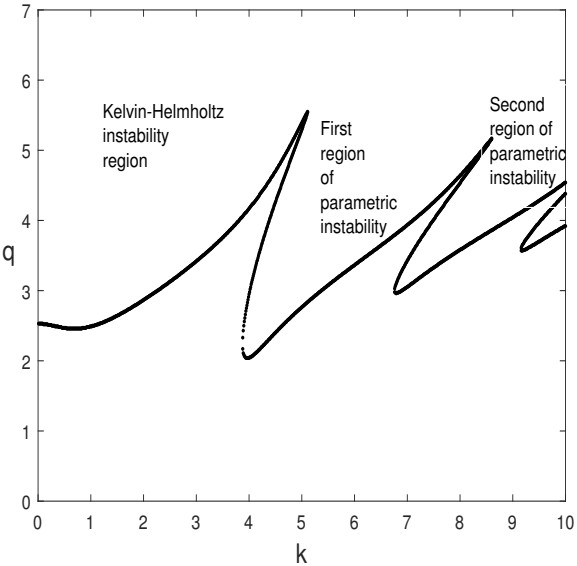

**Figure 2.** Marginal stability curves, $q(k)$, in the periodic case ($A = 0$) for $\rho = 0.8$, $\mu = 0.5$, $F = 0.1$, $We = 5$, $H_1 = H_2 = 2$.

### 4.2. Effect of the Irrational Frequency Ratio in the Case of Equal Amplitudes of Superimposed Accelerations: $a_1\omega_1^2 = a_2\omega_2^2$

In this section, we assume that the two fluids are subjected to superimposed accelerations with equal amplitudes, $a_1\omega_1^2 = a_2\omega_2^2$. Hereafter, we focus attention on the marginal stability curves, $q(k)$ in Figure 3a–f, for different values of the irrational ratio of frequencies, $\Omega = \frac{\omega_1}{\omega_2} = \sqrt{A} = \sqrt{\frac{a_2}{a_1}}$, and for the following assigned values $\rho = 0.8$, $\mu = 0.5$, $F = 0.1$, $We = 5$, and $H_1 = H_2 = 2$. Note that the frequency $\omega_1$ is fixed via the Weber number, $We$, and the frequency number, $\sigma_2$.

#### 4.2.1. Kelvin-Helmholtz Instability

As can be seen in Figure 3a,b, the Kelvin-Helmholtz instability occurs in the long-wavelength regime. Indeed, it occurs at the wavenumber at which the calculation is initiated, $k = 0.01$ for $q = 0.62$ and $q = 1.47$, respectively, for $\Omega = \frac{1}{\sqrt{37}}$ and $\Omega = \frac{1}{\sqrt{2}}$. However, by inspecting the curves in Figure 3c–f, we can see that, this time, the increase in the irrational ratio of frequencies $\Omega$ from $\sqrt{2}$ to $\sqrt{37}$ corresponds to a decrease in the frequency $\omega_2$, which tends to transform the curves of the Kelvin-Helmholtz instability

without a threshold into curves with a threshold with an expansion of their corresponding region. Indeed, the instability curves change their behavior and become curves with a threshold (minimum in amplitude) contrary to Figure 3a,b. For instance, the curve in Figure 3c, corresponding to $\Omega = \sqrt{2}$, has a minimum at $k = 0.63$ corresponding to $q = 2.03$. In Figure 3e, the threshold of this instability corresponds to $q = 2.35$ at $k = 0.82$ for $\Omega = \sqrt{11}$. These results also indicate a tendency to stabilize as $\Omega$ increases. It follows from these findings that the irrational ratio of frequencies has a significant influence on the Kelvin-Helmholtz instability, in the sense that it can change from a non-threshold instability (long wavelength) to a threshold instability (finite wavelength).

### 4.2.2. Parametric Resonances

Besides the influence of $\Omega$ on the Kelvin-Helmholtz instability, and as illustrated in Figure 3a–c for the limiting case of low values of $\Omega$ ($\frac{1}{\sqrt{37}}$, $\frac{1}{\sqrt{2}}$ and $\sqrt{2}$), the quasi-periodic oscillation gives rise to more resonance zones than the periodic case [13].

For $\Omega = \frac{1}{\sqrt{37}}$, the first parametric instability, that occurs at $k = 1.63$ for $q = 0.6$, corresponds to the most unstable mode. For $\Omega \in [\frac{1}{\sqrt{2}}, \sqrt{3}]$, the increase in $\Omega$ shifted the most unstable resonance to the short wavelength perturbations. It takes place at $k = 4$ for $\Omega = \frac{1}{\sqrt{2}}$, at $k = 4.43$ for $\Omega = \sqrt{2}$, and at $k = 4.76$ for $\sqrt{3}$. Moreover, for $\Omega = \sqrt{11}$, the sixth resonance is the most unstable one and its threshold corresponds to $k = 4$ and $q = 2$. As can be seen, the increase in $\Omega$ is accompanied by a decrease in the dimensionless critical amplitude which corresponds to a destabilizing effect. In addition, it can be seen in Figure 3d–f that some parametric instability zones start to disappear by increasing this parameter beyond $\sqrt{3}$. From $\Omega = \sqrt{37}$, the marginal stability curves become similar to the periodic case and the most unstable resonance is the first one at $k = 4.01$ and $q = 2.01$, as illustrated in Figure 3f.

As a summary, the selection of the wavenumber strongly depends on the irrational ratio of frequencies, $\Omega$. For low values of this ratio, the most unstable resonance zones occur for low wavenumbers (long wavelengths). The wavenumber increases with $\Omega$ with a stabilizing effect and beyond $\Omega = \sqrt{37}$, the results tend towards those of periodic oscillation [13] where the most dangerous parametric mode is the first one occurring at $k = 4$. Note that, this behavior was also observed in the literature [17–19]. In other words, it should be noted that, the flow dynamics observed in the case of high values of $\Omega$ ($\omega_2 \longrightarrow 0$) is similar to that in the periodic oscillation case, where only the acceleration relative to the frequency $\omega_1$ is dominant.

To better highlight the quasi-periodicity at the threshold of the parametric instability (resonance), we refer to the method of harmonic balances [17,18], which consists of inserting the expression $\xi(\tau) = \sum_{n=0}^{\infty} \sum_{m=-\infty}^{\infty} \left[ C_{nm} \cos\left(\frac{n+m\Omega}{2}\tau\right) + D_{nm} \sin\left(\frac{n+m\Omega}{2}\tau\right) \right]$ into Equation (27). Results are obtained by a truncation of the infinite sums in this expression and then replaced by sums from 0 to $N$ for $n$ and from $-N$ to $N$ for $m$, respectively, which allows us to obtain two coupled homogenous algebraic systems in $C_{nm}$ and $D_{nm}$ which verify $C_{-n,-m} = C_{n,m}$ and $D_{-n,-m} = -D_{n,m}$. The system has a non-trivial solution if its determinant vanishes. For each $N$, the dimension of this system is $2N^2 + 2N + 1$. In this context, in Figure 4, we illustrate the frequency spectrum of solutions generated at the minimum of two resonances of Figure 3c. Figure 4 shows the largest amplitude corresponding to the dominant mode. Each dominant mode can be identified in the double series we have just presented. Indeed, the numerical procedure shows that the thresholds of the two resonances correspond, respectively, to the modes ($n = 1, m = -1$) and ($n = 1, m = 1$).

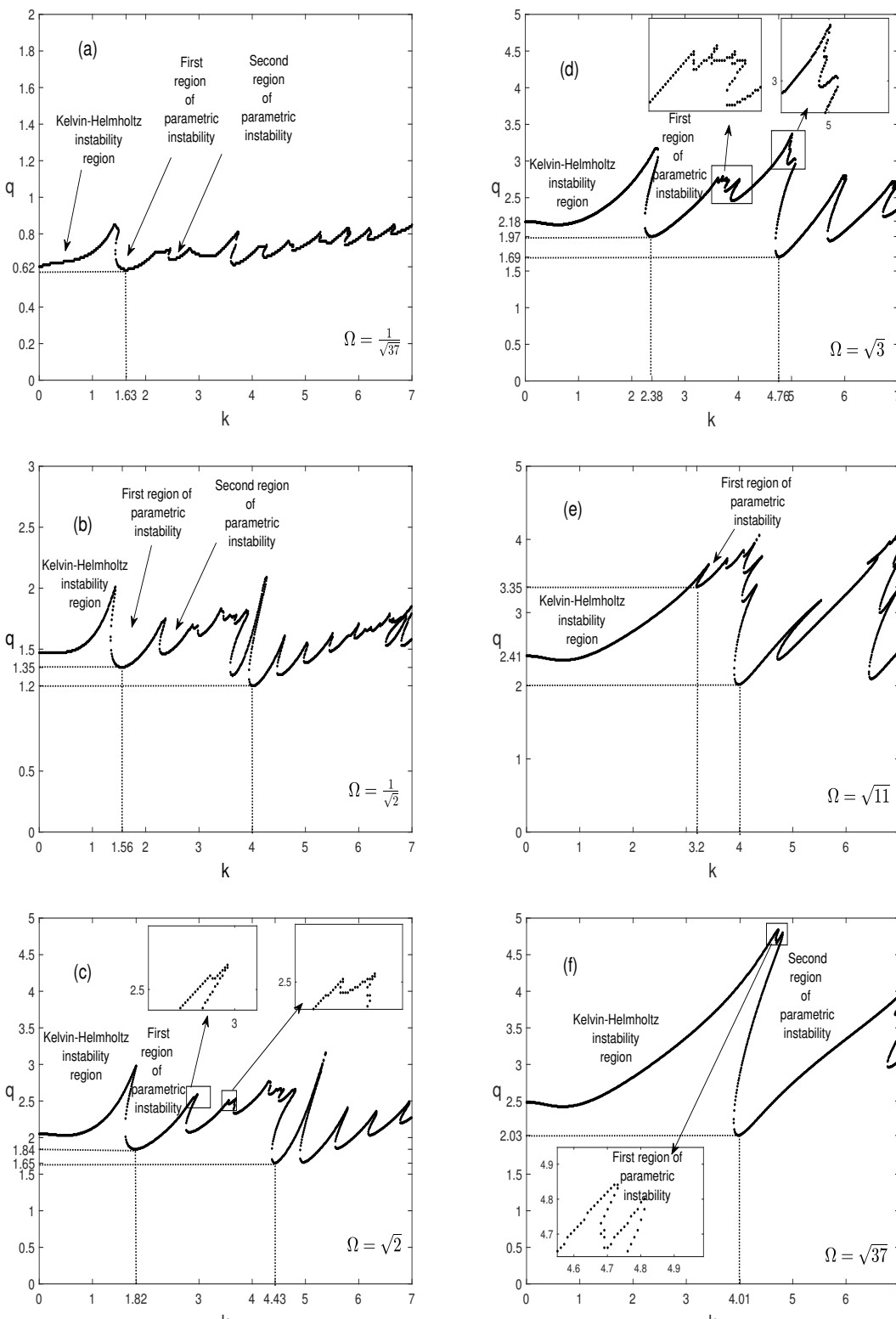

**Figure 3.** Marginal stability curves, $q(k)$, in the case of equal superimposed accelerations $a_1\omega_1^2 = a_2\omega_2^2$, for different values of the irrational ratio of frequencies, $\Omega$, and for $\rho = 0.8$, $\mu = 0.5$, $F = 0.1$, $We = 5$, $H_1 = H_2 = 2$.

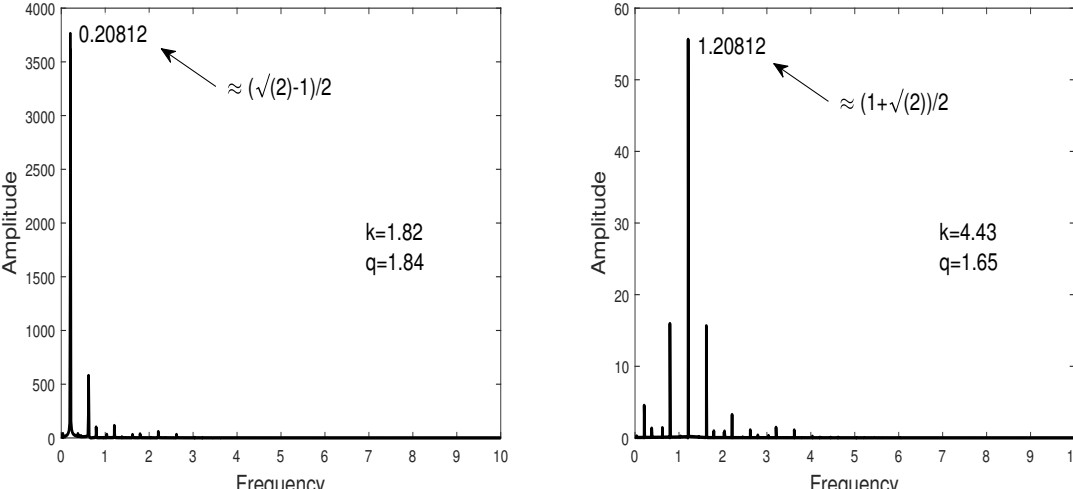

**Figure 4.** Frequency spectrum at the threshold resonances located at $k = 1.82$ and $k = 4.43$ for $\Omega = \sqrt{2}$.

*4.3. Effect of the Irrational Frequency Ratio in the Case of Equal Amplitudes of Superimposed Displacements, $a_1 = a_2$*

In this context, we assume equal amplitudes of the superimposed quasi-periodic oscillations ($A = \frac{a_2}{a_1} = 1$). In Figure 5a–f, we report the marginal stability curves corresponding to the influence of the irrational ratio of frequencies, $\Omega$, on the instability regions for $\rho = 0.8$, $\mu = 0.5$, $F = 0.1$, $We = 5$, and $H_1 = H_2 = 2$.

For $\Omega \longrightarrow 0$ ($\omega_2 \longrightarrow \infty$), in Figure 5a, corresponding to $\Omega = \frac{1}{\sqrt{37}}$, four resonances are detected in the interval of the wavenumber $0 \leq k \leq 7$. In this situation, the most unstable region corresponds to the second parametric instability that occurs at $k = 3.99$ for $q = 2.02$. However, the Kelvin-Helmholtz instability curve has a minimum at $k = 0.68$ and $q = 2.44$.

The passage of curves in Figure 5a, for $\Omega = \frac{1}{\sqrt{37}}$, to the ones in Figure 5b, corresponding to $\Omega = \frac{1}{\sqrt{2}}$, is accompanied by the appearance of other resonance zones. Note that, the eighth one is the most unstable, showing a destabilizing effect. It occurs at $k = 3.98$ and $q = 1.63$. Furthermore, the region of the Kelvin-Helmholtz instability is narrowed and its threshold is decreased.

By increasing $\Omega$ from $\frac{1}{\sqrt{2}}$ to $\sqrt{2}$, in Figure 5c, we notice that the parametric instability zones shift to the right and the ninth resonance is the most unstable, occurring at $k = 4.91$ for $q = 1.11$, and always with a destabilizing effect. In addition, a new weak expansion of the Kelvin-Helmholtz instability region is also observed with a destabilizing effect. The results in Figure 5d, corresponding to $\Omega = \sqrt{3}$, show that the thresholds of the two types of instability continue to decrease and the Kelvin-Helmholtz instability region is expanding. Furthermore, the resonances are displaced toward higher wavenumbers and the most unstable resonance becomes the sixth resonance taking place at $k = 5.53$ and $q = 0.95$. The increase in $\Omega$ from $\sqrt{3}$ to $\sqrt{37}$, as shown in Figure 5e,f, tends to suppress the parametric resonances from the wavenumber interval under consideration and only the Kelvin-Helmholtz instability persists and occurs for a smaller dimensionless amplitude of oscillation.

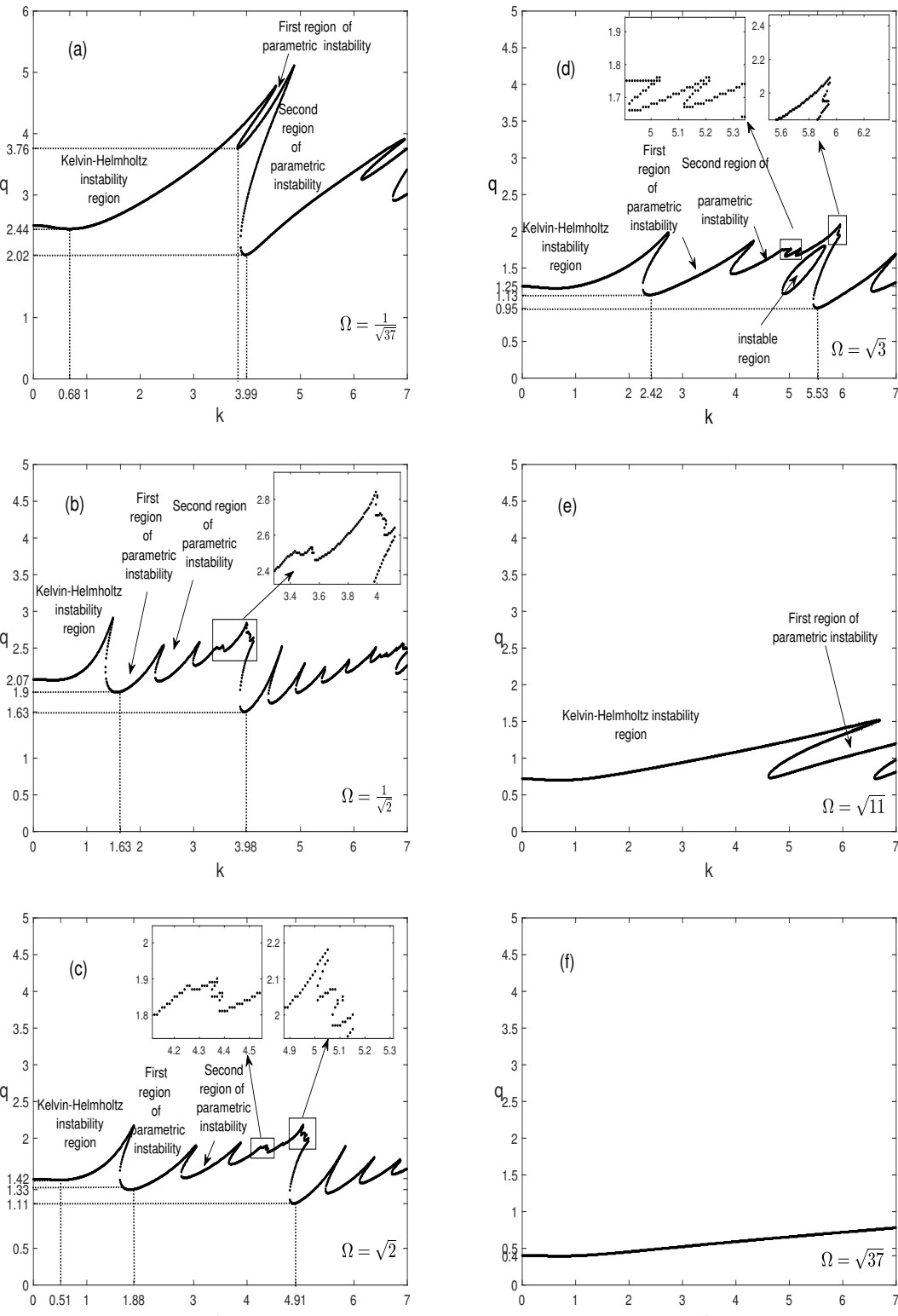

**Figure 5.** Marginal stability curves, $q(k)$, in the case of equal amplitudes of the superimposed displacements, $a_1 = a_2$ ($A = 1$), for different values of the irrational ratio of frequencies, $\Omega$, and for $\rho = 0.8$, $\mu = 0.5$, $F = 0.1$, $We = 5$, $H_1 = H_2 = 2$.

### 4.4. Effect of the Dimensionless Amplitude of Oscillation A for Different Irrational Ratio of Frequencies Ω

The effect of the dimensionless amplitude of oscillation, $A = \frac{a_2}{a_1}$, on the Kelvin-Helmholtz and the parametric instabilities is considered in Figures 6 and 7 for different irrational ratio of frequencies, Ω.

For $\Omega = \frac{1}{\sqrt{37}}$, inspecting the curves in Figure 6a–c, it can be seen that, for $A = 0.1$, the curves of Figure 6a correspond to the periodic oscillation [13] as in Figure 2, and the increase in the oscillation amplitude, from $A = 0.1$ to $A = 1$, gives rise to further regions of parametric instability in the wavenumber interval $0 \le k \le 7$ without a significant effect on the thresholds. Indeed, the thresholds of the most dangerous modes and those of the Kelvin-Helmholtz instability region are not affected.

On the other hand, in contrast to the value $\Omega = \frac{1}{\sqrt{37}}$, $\Omega = \sqrt{3}$ promotes the occurrence of more resonances into the marginal stability curves in Figure 6d–f. Furthermore, it turns out that the increase in the amplitude, $A$, has a destabilizing effect and tends to decrease the thresholds of the Kelvin-Helmholtz instability and those of the parametric resonances. Let us also note that the latter are slightly shifted towards the short wavelength region.

Figure 7 shows that, for a large value of the irrational ratio of frequencies, $\Omega = \sqrt{37}$, the increase in $A$ acts to destabilize the Kelvin-Helmholtz instability and suppresses the parametric instability (resonances) in the range of the wavenumber $0 \le k \le 7$.

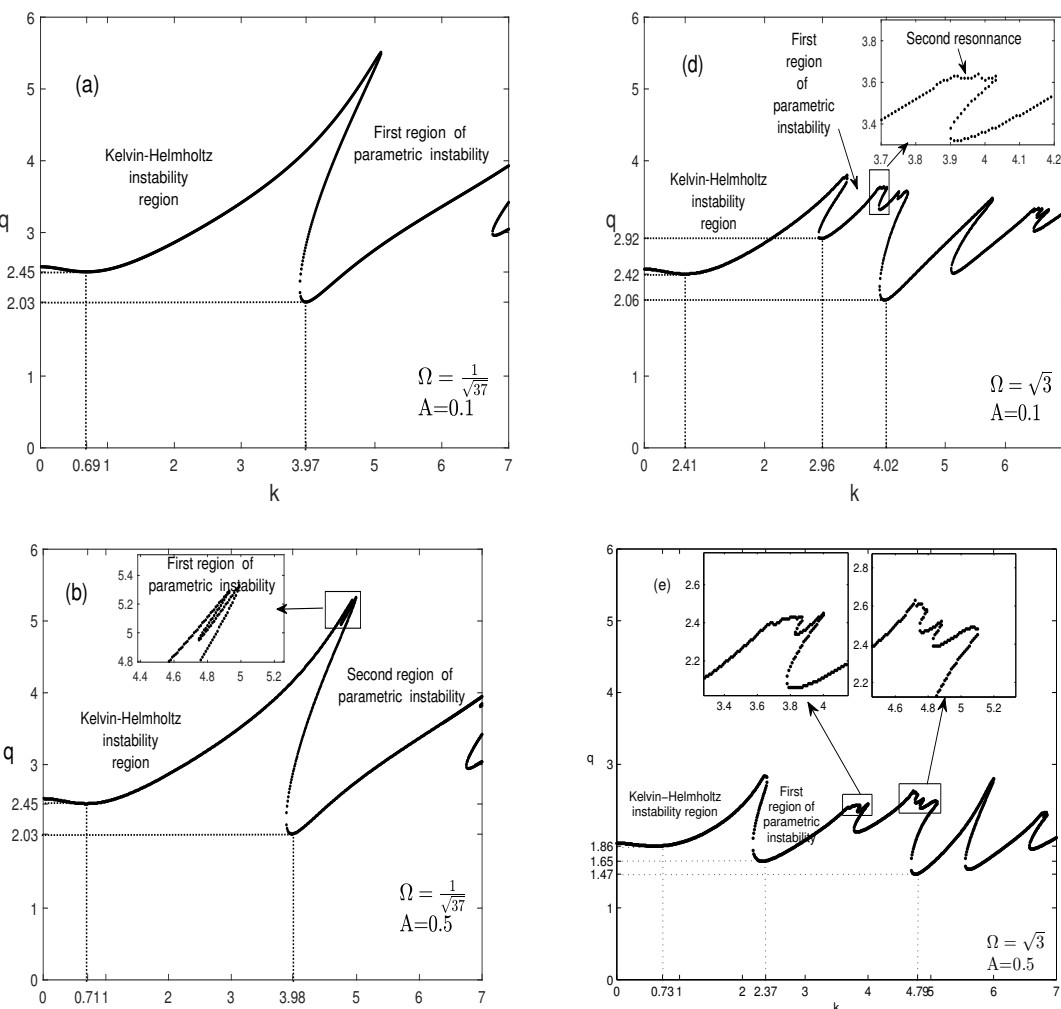

**Figure 6.** *Cont.*

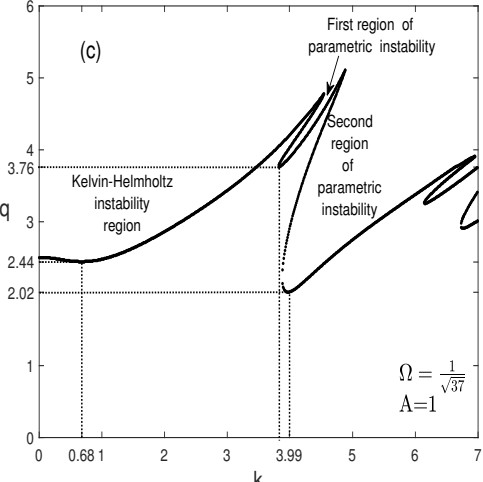
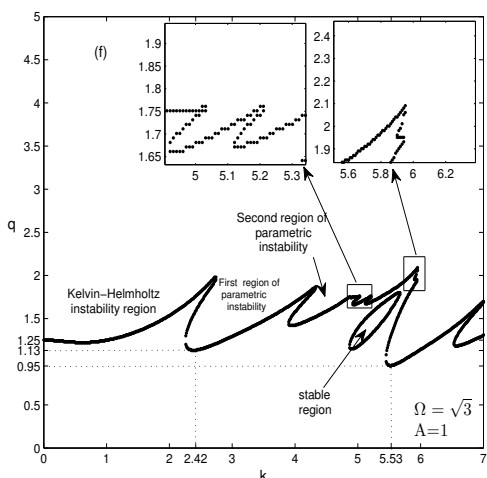

**Figure 6.** Marginal stability curves, $q(k)$, for $\Omega = \frac{1}{\sqrt{37}}$, $\Omega = \sqrt{3}$ and for different values of $A$ (ratio of the amplitudes of the superimposed displacements), $\rho = 0.8$, $\mu = 0.5$, $F = 0.1$, $We = 5$, $H_1 = H_2 = 2$.

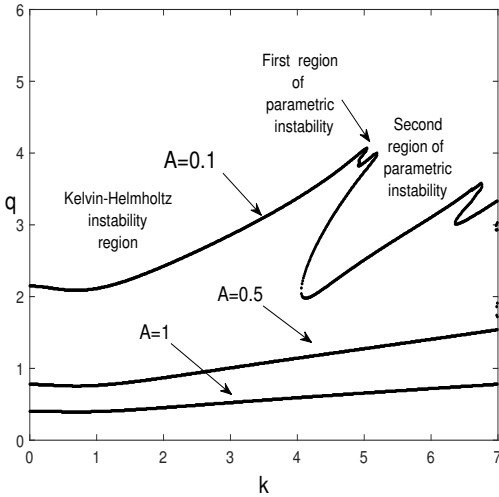

**Figure 7.** Marginal stability curves, $q(k)$, for $\Omega = \sqrt{37}$ and for different values of the amplitude, $A$ (ratio of the amplitudes of the superimposed displacements), $\rho = 0.8$, $\mu = 0.5$, $F = 0.1$, $We = 5$, $H_1 = H_2 = 2$.

### 4.5. Effect of the Damping Coefficient F

To examine the influence of the damping coefficient responsible for the friction, $F = \frac{\sqrt{\frac{l_c}{g}}}{\sigma_2}$, the neutral curves, $q(k)$, are presented for different values of $F$, and for the following assigned values: $A = 1$, $\rho = 0.8$, $\mu = 0.5$, $We = 5$, and $H_1 = H_2 = 2$.

The curves in Figure 8 are plotted in the pertinent case of a low irrational ratio of frequencies, $\Omega = \frac{1}{\sqrt{37}}$. As expected, the increase in the parameter $F$, corresponding to a decrease in the frequency number $\sigma_2$, and thus an increase in the viscosity of the upper fluid $\nu_2$, tends to systematically increase the threshold of the Kelvin-Helmholtz instability region and that of the resonances. In addition, the parametric instability regions are shifted into those of the short wavelength in which the resonances can be suppressed by viscosity. Note that, when $F = 0$ ($\sigma_2 \to \infty$), the results tend towards those of Khenner et al. [5], corresponding to the inviscid case.

*4.6. Effect of the Weber Number, We, on the Stability Threshold*

The influence of the Weber number, *We*, on the neutral curves $q(k)$ is presented, in Figure 9, for the relevant case of the small irrational ratio of frequencies $\Omega = \frac{1}{\sqrt{37}}$, $A = 1$ and for $We = 1$, 5, and 13. As can be seen, an increase in the Weber number, corresponding to an increase in the oscillation frequency, $\omega_1$, leads to a considerable downward shift in the marginal stability curves, indicating the occurrence of a destabilizing effect with a remarkable expansion of the Kelvin-Helmholtz instability region. It should also be noted that, on the one hand, the increase in *We* shifts the most unstable resonance zone towards short wavelength perturbations (large wavenumber), whilst on the other hand, it tends to reduce the number of resonance zones in the wavenumber interval considered in this study.

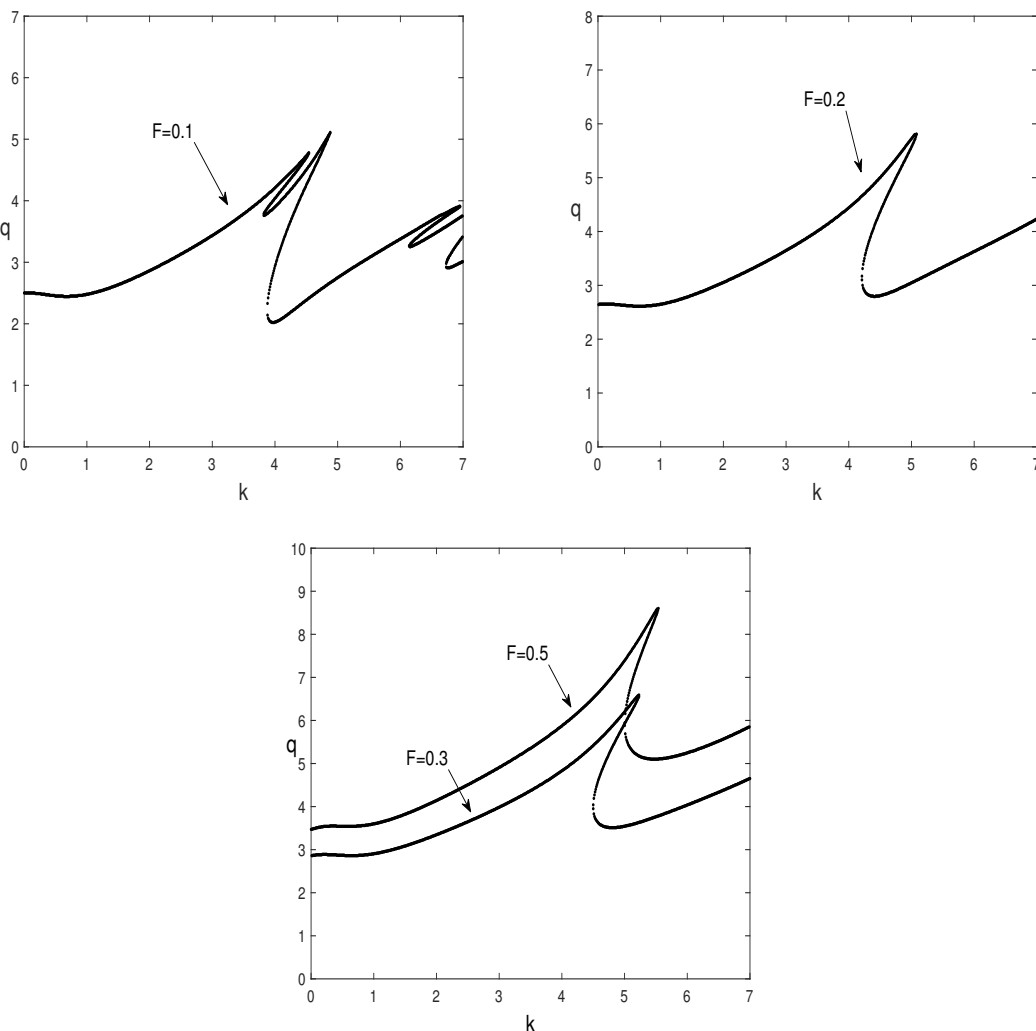

**Figure 8.** Marginal stability curves, $q(k)$, for different values of the damping coefficient, *F*, responsible for the friction, $\rho = 0.8$, $\mu = 0.5$, $We = 5$, $H_1 = H_2 = 2$, $\Omega = \frac{1}{\sqrt{37}}$, $A = 1$.

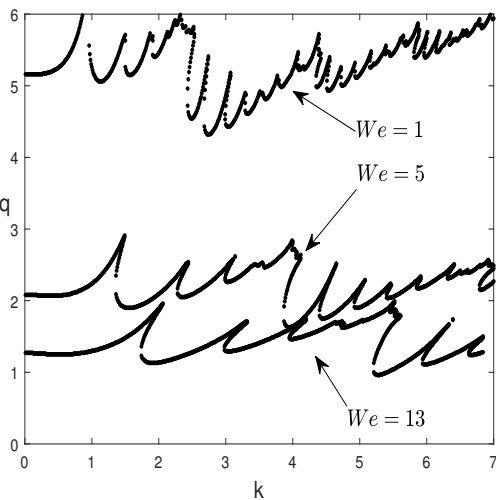

**Figure 9.** Marginal stability curves, $q(k)$, for different values of the Weber number, $We$, and for $\Omega = \frac{1}{\sqrt{37}}$, $A = 1$, $\rho = 0.8$, $\mu = 0.5$, $F = 0.1$, $H_1 = H_2 = 2$.

These results are also presented under another aspect in Figure 10, which represents the evolution of the thresholds of the Kelvin-Helmholtz instability, $(q_{kh}, k_{kh})$ and of the most unstable resonance, $(q_R, k_R)$, as a function of the Weber number, $We$. We notice that the critical wavenumber, $k_{kh}$, slightly increases from almost zero, with the increase in $We$, which means that the instability without the threshold becomes with threshold. However, the corresponding critical amplitude $q_{kh}$ decreases with $We$, thus showing a destabilizing effect. Likewise, the increase in $We$ decreases the critical amplitude, $q_R$, of the first parametric instability and increases the corresponding wavenumber.

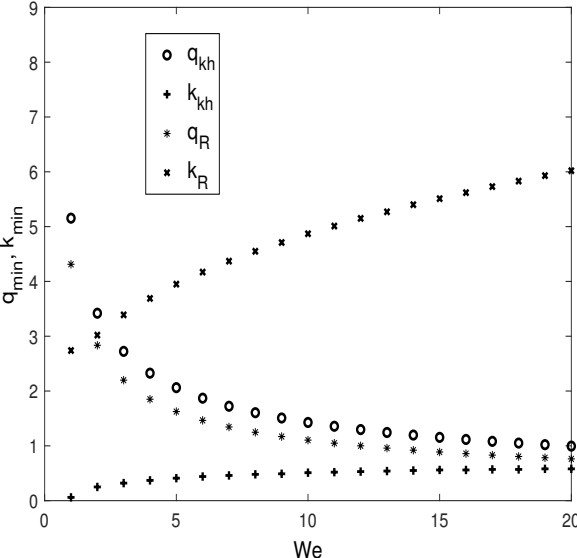

**Figure 10.** Dependence of the thresholds of the Kelvin-Helmholtz instability, $(q_{kh}, k_{kh})$, and of the first resonance, $(q_R, k_R)$, on the Weber number, $We$, for a small irrational ratio of frequencies, $\Omega = \frac{1}{\sqrt{37}}$ and for $A = 1$, $\rho = 0.8$, $\mu = 0.5$, $F = 0.1$, $H_1 = H_2 = 2$.

### 4.7. Effect of the Density Ratio $\rho$

The effect of the density ratio, $\rho$, on the Kelvin-Helmholtz and parametric instabilities, is considered in Figure 11a–c. The configuration is initially stable $(\rho_1 > \rho_2)$, hence the density ratios are chosen to be less than unity. In Figure 11a, it is found that, in the interval $(0.1 \le \rho \le 0.4)$, an increase in the density ratio decreases the threshold values of the most

unstable resonance, which corresponds to the first one as well as the threshold values of the Kelvin-Helmholtz instability whose zone widens.

Afterwards, for ($\rho > 0.4$), Figure 11b,c show that the neutral curves of the Kelvin-Helmholtz instability and the resonances are located above the ones corresponding to $\rho = 0.4$. Here, as can be observed in Figure 12, for $\rho \le 0.4$, increasing the ratio $\rho$ decreases the thresholds while, for $\rho > 0.4$, an opposite effect is observed. This change from a destabilizing effect to a stabilizing one occurs for $\rho = 0.4$, where the threshold of the Kelvin-Helmholtz instability, as can be seen in Figure 11b, corresponds to the critical amplitude and wavenumber $q = 1.4$ and $k = 0.65$, respectively. Thus, on the one hand, for $\rho = 0.9$, the results in Figure 11c show that the threshold significantly increases to reach the critical parameters $q = 3.514$ and $k = 0.7$; on the other hand, only one resonance exists in the considered interval of the wavenumber. Let us also note that, in Figure 11b,c, the first resonance dominates and the competition between the other parametric modes is suppressed, especially for $\rho = 0.9$. These results are also illustrated in Figure 12 where we observe, for $\rho > 0.4$, that the thresholds of the Kelvin-Helmholtz instability and the first resonance, $q_{kh}$ and $q_R$, grow monotonically with the increase in $\rho$. Nevertheless, the wavenumber of the first resonance significantly increases while that of the Kelvin Helmholtz instability remains almost constant.

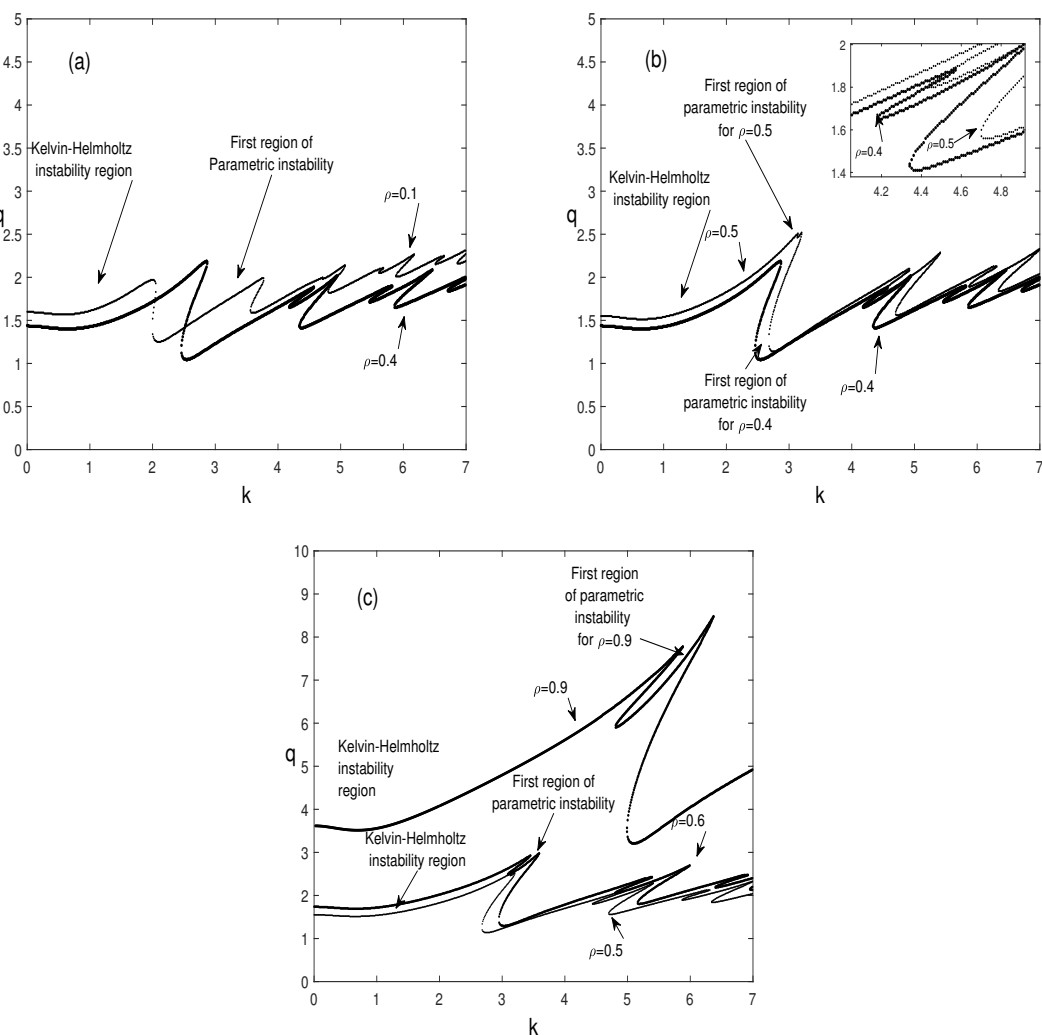

**Figure 11.** Marginal stability curves, $q(k)$, for different values of the density ratio, $\rho$, and for $\Omega = \frac{1}{\sqrt{37}}$, $A = 1$, $\mu = 0.5$, $F = 0.1$, $We = 5$, $H_1 = H_2 = 2$.

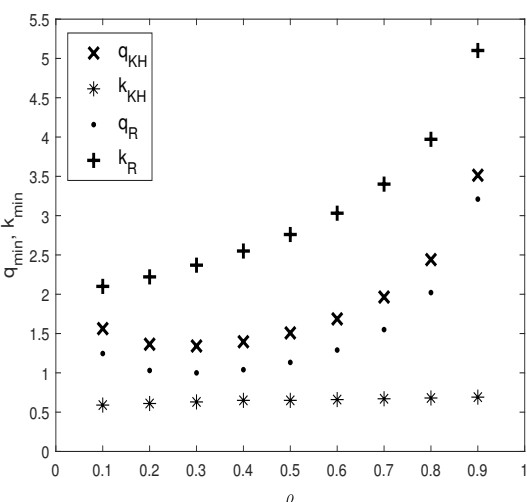

**Figure 12.** Dependence of the thresholds of Kelvin-Helmholtz instability, $q_{KH}$ and $k_{KH}$, and of the first resonance, $q_R$ and $k_R$, on the density ratio, $\rho$, for $\Omega = \frac{1}{\sqrt{37}}$, $A = 1$, $\mu = 0.5$, $F = 0.1$, $We = 5$, $H_1 = H_2 = 2$.

*4.8. Effect of the Heights of the Two Fluid Layers $H_1$ and $H_2$*

In this section, the effect of the dimensionless thicknesses of the fluid layers $H_1$ and $H_2$ on the stability of the interface is illustrated in Figure 13, assuming that the layers have the same thicknesses, i.e., $H = 1$. It is worth noting that the increase in $H_1$ and $H_2$ has no effect on the resonance zones. In addition, by increasing $H_1$ and $H_2$, with $H_1 = H_2 > 1$, the threshold of the Kelvin Helmholtz instability, initially with a very large wavelength, has a finite wavelength. This behavior was also observed by Khenner et al. [5] and Lyubimova et al. [13] for a periodic horizontal oscillation.

Figure 14 illustrates the evolution of the dimensionless amplitude of oscillation, $q$, versus the wavenumber, $k$, for different values of $H$ ($H \neq 1$) and for the same other parameters as in Figure 13. It turns out that increasing $H$ has a stabilizing effect on the Kelvin-Helmholtz and parametric instabilities.

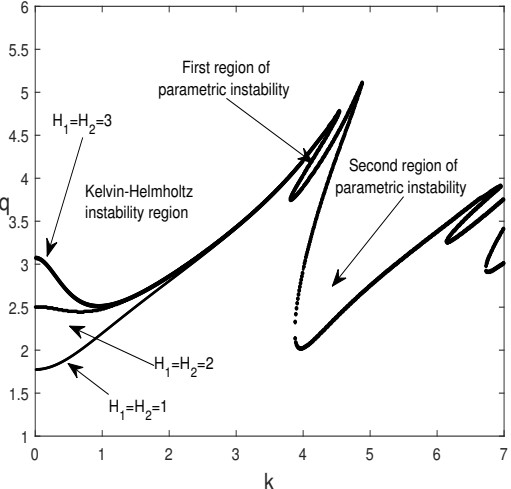

**Figure 13.** Marginal stability curves, $q(k)$, for $\Omega = \frac{1}{\sqrt{37}}$, $\rho = 0.8$, $\mu = 0.5$, $F = 0.1$, $We = 5$, $A = 1$ and for different values of $H_1$ and $H_2$ with $H = 1$.

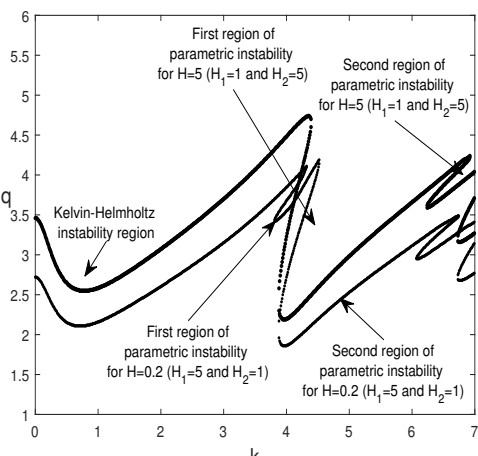

**Figure 14.** Marginal stability curves, $q(k)$, for $\Omega = \frac{1}{\sqrt{37}}$, $\rho = 0.8$, $\mu = 0.5$, $F = 0.1$, $We = 5$, and $A = 1$, and for different values of $H_1$ and $H_2$ with $H \neq 1$.

## 5. Conclusions

In this study, we conducted a linear stability analysis of a system of two superimposed fluid layers, of different densities and viscosities, confined in a vertical Hele-Shaw cell subjected to horizontal quasi-periodic oscillation. The linear problem was reduced to a quasi-periodic oscillator governing the evolution of the amplitude of the interface separating the two fluid layers. In this investigation, we focused our analysis on the dependency of the marginal stability curves on the irrational ratio of frequencies, $\Omega$, for equal amplitudes of the superimposed accelerations of the quasi-periodic oscillation, $a_1\omega_1^2 = a_2\omega_2^2$, and for different ratios of the superimposed displacement amplitudes, $A = 1$ ($a_1 = a_2$), $A = 0, 1$, and $A = 0, 5$. The effects of the damping coefficient, $F$, the Weber number, $We$, the density ratio, $\rho$, and the dimensionless heights of the two fluid layers $H_1$ and $H_2$ were also discussed.

For equal superimposed acceleration amplitudes, $a_1\omega_1^2 = a_2\omega_2^2$, the increase in the irrational ratio of the frequencies $\Omega = \frac{\omega_1}{\omega_2}$ renders the Kelvin-Helmholtz instability initially without threshold into an instability with a threshold. The numerical results have also shown that for low-frequency ratios, there is a rich dynamics in terms of the resonance number, allowing a very large selection of the wavenumber which results in the control of the wave size. For both types of instability, increasing the irrational frequency ratio has a stabilizing effect with the disappearance of several resonances, and the marginal stability curves evolve towards the periodic case.

In the case of equal amplitudes' displacements, $A = 1$, the very low values of the irrational ratio of frequencies, $\Omega$, this time give results that are similar to the periodic case, whereas the intermediate values of this ratio give rise to several resonances with a destabilizing effect for the two types of instability when increasing $\Omega$. For large ratios, the parametric resonances are suppressed from the considered wavenumber interval. However, for $A = 0.5$, $A = 0.1$, and for low values of $\Omega$, the dimensionless amplitude, $A$, has no effect on the Kelvin-Helmholtz instability and on the parametric resonances thresholds. However, for an intermediate value of $\Omega$, it has a destabilizing effect on the two types of instability with the presence of several resonances.

For the effect of the damping coefficient responsible for the friction $F$ and the Weber number, the numerical results showed that increasing these two dimensionless numbers has a stabilizing effect on the Kelvin-Helmholtz instability, and that the parametric instabilities are shifted to the short wavelength as the irrational frequency ratio increases.

Concerning the effect of the density ratio, for $\rho \leq 0.4$, it has a destabilizing effect on the Kelvin-Helmholtz and the parametric instabilities. However, for $\rho \geq 0.4$, the density ratio has a re-stabilizing effect on the two types of instability.

Finally, keeping $H = 1$, and increasing the heights of the two fluid layers has no effect on the resonances but has a stabilizing effect on the Kelvin-Helmholtz instability, which was initially thresholdless but becomes so.

**Author Contributions:** Conceptualization, S.A. and J.B.; investigation, M.A., A.E.J. and M.E; visualization and writing-original draft, J.B.; review and editing, S.A.; supervision, S.A. All authors have read and agreed to the published version of the manuscript.

**Funding:** This research was supported by the Hassan II University of Casablanca.

**Conflicts of Interest:** The authors declare no conflict of interest.

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
