# Peer review of "Effect of Horizontal Quasi-Periodic Oscillation on the Interfacial Instability of Two Superimposed Viscous Fluid Layers in a Vertical Hele-Shaw Cell"

_fluids, doi:10.3390/fluids8060164_

Round 1
Reviewer 1 Report
In this paper, the authors report the results of ''Effect of horizontal quasi-periodic oscillation on the interfacial instability of two superimposed viscous fluid layers in a vertical Hele-Shaw cell'' I enlist them below:
1. Some grammatical and typo mistakes are found in the manuscript. carefully rectify all such mistakes in the entire manuscript.
2. Please consider citing some literature relevant to ''quasi-periodic oscillation'' in introduction section.
3. Cite references to equations e.g eq (2).
4. Add Nomenclature as some parameters are not clear e.g Λ2 and Λ1
5. Abstract should be improved by adding relevant results
6. Research gap is missing in introduction section and Novelty should be clearly mentioned.
7. In result and discussion section physical significance is required to explain results,
8. Conclusion section must be improved
Author Response
Responses to Reviewer 1
- Some grammatical and typo mistakes are found in the manuscript. carefully rectify all such mistakes in the entire manuscript.
Grammar and typo mistakes are carefully corrected
- Please consider citing some literature relevant to ''quasi-periodic oscillation'' in introduction section.
The following paragraph has been added to the Introduction section
Inspired by the work of Rand et al. who studied the quasi-periodic Mathieu equation, Boulal et al. 16, 17 used this type of modulation in Rayleigh-Bénard convection where it is shown that the ratio of frequencies allows to control the convection threshold. This type of modulation was also used by Yagoubi et al.18 to study the effect of vertical quasi-periodic oscillation on the stability of the free surface of a horizontal liquid layer18.
- Cite references to equations e.g eq (2).
This reference is added to the manuscript
- Add Nomenclature as some parameters are not clear e.g Λ2 and Λ1
All parameters are clarified in the manuscript
- Abstract should be improved by adding relevant results
The following paragraph has been added to the Introduction section:
“whose thresholds are found to correspond to quasi-periodic solutions using the frequency spectrum”
- Research gap is missing in introduction section and Novelty should be clearly mentioned.
The novelty in this work is the control of the wave size, which is widely commented in the abstract and in the conclusion.
- In result and discussion section physical significance is required to explain results,
In our opinion we have discussed extensively the two important physical phenomena in this study which are the Kelvin-Helmholtz instability and the parametric resonances
- Conclusion section must be improved
The conclusion is already quite long and we have made sure that it includes all the results we have obtained

Reviewer 2 Report
This manuscript has investigated a linear stability analysis of a system of two superimposed fluid layers, of different densities and viscosities, conned in a vertical Hele-Shaw cell subjected to horizontal quasi-periodic oscillation. The authors focused on the dependency of the marginal stability curves on the irrational ratio of frequencies for equal amplitudes of the superimposed accelerations of the quasi-periodic oscillation and for different ratios of the superimposed displacement amplitudes. Although the analysis is systematic and adequate, how the results of this study differ from those obtained using the rational ratio of frequencies is unclear. If they are qualitatively different, the points should be clarified.
Author Response
Responses to Reviewer 2
This manuscript has investigated a linear stability analysis of a system of two superimposed fluid layers, of different densities and viscosities, conned in a vertical Hele-Shaw cell subjected to horizontal quasi-periodic oscillation. The authors focused on the dependency of the marginal stability curves on the irrational ratio of frequencies for equal amplitudes of the superimposed accelerations of the quasi-periodic oscillation and for different ratios of the superimposed displacement amplitudes. Although the analysis is systematic and adequate, how the results of this study differ from those obtained using the rational ratio of frequencies is unclear. If they are qualitatively different, the points should be clarified.
Response:
The frequency ratio is approximated by a rational number which has very particular coefficients (page 9). Thus, in subsection B. 2 on parametric resonances, we have identified the frequency corresponding to the instability threshold. This frequency is very close to an irrational frequency.
The following paragraph has been added to the abstract section:
“whose thresholds are found to correspond to quasi-periodic solutions using the frequency spectrum”
